# ZanzaMapp: A Scalable Citizen Science Tool to Monitor Perception of Mosquito Abundance and Nuisance in Italy and Beyond

**DOI:** 10.3390/ijerph17217872

**Published:** 2020-10-27

**Authors:** Beniamino Caputo, Mattia Manica, Federico Filipponi, Marta Blangiardo, Pietro Cobre, Luca Delucchi, Carlo Maria De Marco, Luca Iesu, Paola Morano, Valeria Petrella, Marco Salvemini, Cesare Bianchi, Alessandra della Torre

**Affiliations:** 1Department of Public Health & Infectious Diseases, Laboratory affiliated to Istituto Pasteur Italia—Fondazione Cenci Bolognetti, Sapienza University of Rome, 00185 Rome, Italy; beniamino.caputo@uniroma1.it (B.C.); mattia.manica@protonmail.ch (M.M.); pietrocobre@gmail.com (P.C.); carlooo92@gmail.com (C.M.D.M.); 2Department of Biodiversity and Molecular Ecology, Research and Innovation Centre, Fondazione Edmund Mach, 38010 San Michele all’Adige, Italy; luca.delucchi@fmach.it; 3Institute for Environmental Protection and Research (ISPRA), via Vitaliano Brancati 48, 00144 Roma, Italy; federico.filipponi@gmail.com; 4MRC Centre for Environment and Health, Department of Epidemiology and Biostatistics, School of Public Health, Faculty of Medicine, Imperial College London, London W2 1PG, UK; m.blangiardo@imperial.ac.uk; 5Department of Biology, University of Naples Federico II, 80126 Naples, Italy; dr.l.iesu@gmail.com (L.I.); valeria.petrella@unina.it (V.P.); marco.salvemini@unina.it (M.S.); 6GH s.r.l., via Petralia Sottana, 11, 00132 Rome, Italy; paolamorano72@gmail.com (P.M.); cesare@cesarebianchi.com (C.B.)

**Keywords:** citizen science, mosquito, nuisance, biting activity, tiger mosquito, *Aedes albopictus*

## Abstract

Mosquitoes represent a considerable nuisance and are actual/potential vectors of human diseases in Europe. Costly and labour-intensive entomological monitoring is needed to correct planning of interventions aimed at reducing nuisance and the risk of pathogen transmission. The widespread availability of mobile phones and of massive Internet connections opens the way to the contribution of citizen in complementing entomological monitoring. ZanzaMapp is the first mobile “mosquito” application for smartphones specifically designed to assess citizens’ perception of mosquito abundance and nuisance in Italy. Differently from other applications targeting mosquitoes, ZanzaMapp prioritizes the number of records over their scientific authentication by requesting users to answer four simple questions on perceived mosquito presence/abundance/nuisance and geo-localizing the records. The paper analyses 36,867 ZanzaMapp records sent by 13,669 devices from 2016 to 2018 and discusses the results with reference to either citizens’ exploitation and appreciation of the app and to the consistency of the results obtained with the known biology of main mosquito species in Italy. In addition, we provide a first small-scale validation of ZanzaMapp data as predictors of *Aedes albopictus* biting females and examples of spatial analyses and maps which could be exploited by public institutions and administrations involved in mosquito and mosquito-borne pathogen monitoring and control.

## 1. Introduction

Mosquitoes are among the deadliest animals worldwide due to their capacity to transmit pathogens (e.g., malaria parasites and arboviruses, such as yellow fever, dengue and Zika) affecting hundreds of million people and causing more than 700,000 deaths/year, primarily in tropical regions. As vaccines for most mosquito-borne diseases are not yet available, the prevention of this huge burden to human health largely focuses on tackling the mosquito vectors, primarily through large-scale insecticide-based interventions at country/regional level, with very high costs on a global scale.

In temperate regions, however, mosquitoes are mostly perceived as a nuisance, in spite of the fact that mosquito-borne viruses, such as the West Nile virus, are endemic in several European and North American Countries. Moreover, the colonization of these regions by invasive species (typically belonging to the genus *Aedes*), ongoing since the eighties, has led to several cases of autochthonous transmission of exotic arboviruses such as dengue in Europe and USA [1,2], Zika in the USA [3], and to two chikungunya outbreaks with hundred human infections in Italy [4]. Differently from mosquito control interventions aimed at preventing disease transmission in tropical areas, interventions for nuisance (and risk of pathogen transmission) reduction purposes in temperate regions are rarely organized at country/regional level and largely implemented by private pest control companies appointed either by municipalities to treat public areas or by citizen to treat private ones. 

In both cases, the foundations for the planning of appropriate control plans lie on a good knowledge of the mosquito species behavior, spatial distribution and temporal dynamics, as well as on a constant assessment of the field-efficacy of the implemented control measures [5,6,7]. Indeed, large-scale implementation of surveillance and monitoring activities is very challenging and costly, posing serious constraints to both the optimization of control plans and to the assessment of the risk and dynamics of pathogen transmission [8], particularly in temperate regions where public health threats and available resources are limited. The extensive availability of mobile phones and of Internet connections opens the way to the contribution of the citizenry in solving/minimizing this problem [9]. Indeed, in recent years, citizen science data have been reliably exploited for wildlife surveys by successfully linking citizens to academics and produced very relevant outputs [10]. In Europe, citizen science projects exploiting data from mobile phone applications have been proposed to complement mosquito surveillance in several countries [11]. In the Netherlands, Muggenradar has been used as an effective method to clarify the distribution of the biotypes of the night biting urban mosquito—*Culex pipiens*—thanks to the physical collection (by citizen) and identification (by scientists) of specimens from all over the country [12]. In Germany, Mückenatlas has allowed detection of changes in the country mosquito fauna, again based on samples collected from the citizens and mailed to specialists [13]. In Spain, Mosquito Alert has provided a comprehensive picture of the ongoing invasion of the country by the Asian tiger mosquito (*Aedes albopictus*) [9]; it also provided the first record of a second invasive species, *Aedes japonicus* [14], and revealed a clear pattern of passive dispersal by cars [15], thanks to photos sent by citizens to a specialist entomology team. 

ZanzaMapp (manufacturer, city, country) is the first mobile “mosquito” application for smartphones specifically designed to contribute to the assessment of the perception of mosquito abundance and nuisance in Italy. Differently from the other applications mentioned above, ZanzaMapp prioritizes the number of records to their authentication. For this reason, ZanzaMapp 1.1.1 (GH s.r.l., Rome Italy) was designed to require the least possible effort and to offer the smoothest experience to users, who were not asked to send mosquito photos or samples, but only to fill an easy and quick questionnaire. We envisaged that an app with these characteristics could provide valuable data on mosquito spatial and temporal dynamics both on a large-scale (e.g., to build information-based models on the risk of arbovirus transmission) and on a local-scale (e.g., to identify spatio-temporal hot-spots of mosquito abundance on which to focus control interventions or to assess the effectiveness of the interventions), as well as a significant contribution to the education of citizens on mosquito prevention and control. Indeed, both ZanzaMapp and the associated website provide easy access to scientifically validated information on basic mosquito morphology and bionomics, on the pathogens they transmit, and on control methods and personal protection measures. The present paper analyses ZanzaMapp data obtained from 2016 to 2018 and discusses the results with reference to the exploitation and appreciation of the app by citizens, and to the data consistency with the known biology of the main mosquito species in Italy. In addition, we present the results of a first field validation of ZanzaMapp data as predictors of *Ae. albopictus* biting females and provide examples of large- and local-scale spatial representations of perceived mosquito abundance highlighting the potential of ZanzaMapp to contribute to the correct planning of control interventions. Finally, the paper discusses the added value of complementing conventional entomological surveillance/monitoring schemes with citizen science data collected by smart-phone apps.

## 2. Materials and Methods 

### 2.1. ZanzaMapp Design and Release 

ZanzaMapp is the result of a collaborative effort between the Medical Entomology group of the Department of Public Health and Infectious Diseases in SAPIENZA University and GH s.r.l., a start-up of SAPIENZA University. The app was implemented as a HTML Single Page Application, so it runs both in a normal browser, as a web app, and wrapped in a native app (both Android (GH s.r.l., Rome, Italy) and iOS (GH s.r.l., Rome, Italy)) with Cordova. It is freely available in the Android and iOS app stores (currently available in version 1.1.2, which also includes the possibility to upload photographical records) and on the project’s website [16].

The present work focuses on data collected by ZanzaMapp 1.1.1 (workflow shown in Appendix A). Users are requested to fill in a brief questionnaire consisting in 4 multiple choice questions (Q): Q1—number of observed mosquitoes (possible answers: 0, 1–3, 3–30, >30); Q2—the time of the observation (possible answers: within an hour from the record; previous night; morning; afternoon); Q3—the location of the observation (possible answers: indoor vs. outdoor); Q4—observation associated to biting activity (possible answers: no, low, high). Questions 1 and 2 are compulsory (1,2) and are followed by the geo-location of the record. Questions 3 and 4 are optional, in order to minimize the effort requested to the user and likely maximize the number of records. Both the native app and the web app use the geolocation capabilities of the users’ devices to show the location on a map of the device at the moment the record is sent, and a pin is displayed by default in this location. Users are allowed to move the pin to a different location, if needed, as the record could refer to observations in the 24 h before it is sent (see Q2). Anonymized data are collected in a server, and a map including all the reports sent by all users in the previous week (completed with a heat map to show the areas of greater mosquito’s abundance) is made available to all users in the native-app and the web-app.

Both native-app and web-app included sections/folders with (i) in-depth educational materials on mosquito species’ cycle, ecology, and public health relevance and (ii) recent news on mosquitoes and mosquito-borne diseases in Italy and beyond.

The ZanzaMapp release on 4 May 2016 was followed by various advertising initiatives, particularly in weeks 18–19 when it was advertised by interviews in national TV news (RAI, Canale 5, SKY) and radio programmes and in national newspapers. In 2017, following the Chikungunya outbreak in the Lazio region, ZanzaMapp was promoted through a national television scientific dissemination programme (GEO RAI3, week 38) and in national and regional (Lazio) TV newscasts (week 41) and newspapers. In September 2016, ZanzaMapp was promoted in the island of Procida (Naples, Italy) within the framework of field site validation (see Pilot field validation below). In addition, in spring and autumn 2016 and 2017, ZanzaMapp was promoted through dedicated seminaries among students of Parasitology and Zoology at the Faculties of Medicine and of Biological Science of the La Sapienza and Tor Vergata Universities in Rome. No promotion was carried in 2018. Except for the advertisement during the pilot study in Procida and for the mentioning of ZanzaMapp in a local TV interview by the Mayor of Bari at the XXIX Congress of the Italian Society of Parasitology in June 2016, no public administration promoted ZanzaMapp use during the study period.

### 2.2. Descriptive and Statistical Analyses

Descriptive statistics were computed to characterize app usage and user engagement based on ZanzaMapp records provided by citizen scientists across Italy from May 2016 to the end of 2018. In order to optimise the dataset to be analysed, a data cleaning procedure was applied, which removed records with default coordinates (when no GPS service is available the app opens the map at a default location) and records with coordinates outside Italy. 

We collected data on following environmental variables at each record location: (i) elevation (Source: available from ISPRA SINAnet [17]); (ii) imperviousness in a 150 m buffer, corresponding to a percentage of the substitution of the original (semi-) natural land cover or water surface with an artificial, often impervious cover(computed from Imperviousness dataset available from ISPRA SINAnet [17]); (iii) distance from the coastline (computed from coastline available from the Italian National Geoportal, [18]); (iv) distance from inland waters (computed from inland water bodies and river network available from the Italian National Geoportal, [18]). For each environmental variable we obtained a distribution of values and calculated the frequency distribution in each reported mosquito abundance category (0, 1–3, 3–30 and >30 mosquitoes.

The relationship between the number of records and human population density was assessed by linear regression. The number of records and the total population within each Italian province were log-transformed (base10) and used as dependent and independent variables, respectively.

A Generalized Additive Model (GAM) with Binomial distribution was applied to analyse the seasonal dynamics of mosquitoes as monitored by users. The following environmental variables were considered: (i) mean daily temperature, obtained from the gap-free daily MODIS Land Surface Temperature maps, a dataset created by Fondazione E. Mach using the Metz et al. 2014 [19] procedure applied to MODIS LST version 6, with a final resolution of 250 m; the temperature value corresponds to the pixel value at the coordinates of the records. (ii) Human population data obtained from the GSH-POP raster dataset with a spatial resolution of 9 arcsec; in this case, too, the selected valued is the value of the pixel identified by the coordinates of the records. All the data were managed with the GRASS GIS software [20].

The GAM was applied to a subset of data, considering only records flagged as being made in the last hour and during 2017 and 2018, to assess mosquito daily activity only and to avoid the confounding effect of the advertising campaign in the first weeks after the release of ZanzaMapp in June 2016. The records for each location i, day j and hour h (y_ijh_) were categorized as high abundance (>3) or low abundance (≤3) and then modelled as a Bernoulli variable (Y). The logit transformed probability was regressed against year, period (summer, April to October; winter, December to March) and population density with a linear term, while for mean daily temperature (LST), and the interaction between hour of the day and period we considered thin plate splines.
y_ijh_ ~ Bernoulli (p_ijh_),(1)
Logit (p_ijh_) = year_j_ + period_j_ + population_i_ + f (LST_ij_) + f (hours_h_ × period_j_),(2)
E (Y) = p,(3)
Var (Y) = p (1 − p)(4)

The GAM was trained on a subset of the considered data (80%), and the model performance was evaluated on the excluded data (test set).

### 2.3. Management and Spatial Representation Tool

An analytical tool coded in R language was developed in order to allow the simple and automatic processing of spatialized ZanzaMapp records (or of other spatialized mosquito data), analyse the report dataset, create map layouts and export results. The tool was bundled in the R package “CSmosquitoSp” (available at [21]) that makes it possible to combine spatialized mosquito data with spatialized information (presently available for the Italian territory only) corresponding to polygons representing administrative boundaries at different levels (i.e., provinces, municipalities, sub-municipalities, spatial grids) and incorporating information about human population, or alternatively, representing other custom polygon layers. Qualitative information about mosquito abundance, reported by citizen scientists in discrete numeric classes (0, 1–3, 3–30, >30 mosquito), is transformed into a numeric scale representing the median value for each class (0, 2, 17, 34 mosquito). The converted reports can be aggregated for a specified time interval over each spatial polygon using basic statistical methods (i.e., average), and positive records are computed as the percentage of records with a number of observed mosquitos >0. The perceived mosquito abundance is computed as fraction abundance (ranging from 0 to 1) for each polygon, by averaging the numeric transformed reports in a specific time interval. “Fraction abundance” is a fractional value, scaled in the range 0–1, of mosquito abundance. A value of “0” represents a spatial polygon where all records in a specific time interval reported a ZanzaMapp abundance of “0” (None) mosquitoes. A value of “1” represents a spatial polygon where all records in a specific time interval reported a ZanzaMapp abundance of “>30” mosquitoes. The fractional value is computed after converting the class records to class median values and averaging all the records inside each spatial polygon for a specific time interval. When scaling in the range 0–1, the divisor corresponds to the highest-class median value (i.e., 34). The following associated information related to uncertainties can be also computed: (i) reliability, expressed as the coefficient of variation calculated from the reports; (ii) sampling effort, representing the ratio between number of reports and human population. Optionally, the analysis can take into account the time of observation, in order to estimate the perceived abundance during daytime and night-time, i.e., associated to diurnal (e.g., *Ae. albopictus* in urban areas) and nocturnal (e.g., *Cx pipiens* in urban areas) species. Aggregating reports within administrative polygons to calculate the perceived mosquito abundance makes it possible to analyse the spatial and temporal distribution, combine them with the quantitative information about the human population, and support the integration with HLC estimates.

CSmosquitoSp has been employed to create some examples of possible maps obtainable from the ZanzaMapp records (see Conclusion Section). To this aim, qualitative information about mosquito abundance, reported by citizen scientists in discrete numeric classes, is represented as perceived mosquito fraction abundance.

### 2.4. Pilot Field Validation

ZanzaMapp data were compared with the results of conventional entomological monitoring carried out in September 2016 in Procida island, a small (3.7 km^2^), highly touristic Mediterranean island in the gulf of Naples (Italy). The island is largely urbanized and densely inhabited (2833 inhabitants/km^2^—ISTAT 31/12/2018); this population density approximately doubles during the summer months, because of tourism. The typical architecture is characterised by family houses with their own ornamental and vegetable gardens, with plenty of opportunities for *Ae. albopictus* larval development (e.g., buckets filled with rainwater for plant watering). 

An intensive entomological field monitoring was carried out daily from 9 to 15 September, except for 14 September, due to adverse weather conditions, for a total of 6 days of collections. Human Landing Collections (HLCs) were carried out during the hours of highest *Ae. albopictus* activity (9:30–12:30; 14:30–19:30) in 164 randomly selected sampling spots by two teams of two trained people (all co-authors of the present paper). All experiments were performed in accordance with WHO guidelines and Italian regulations. The project was approved by the Municipality of Procida with municipal resolution n°52 of 7 July 2016. Considering the absence of arboviruses spreading in the study area and the methods utilized, no additional permits were required. In each sampling spot, HLC were carried out simultaneously for 5 min by the two collectors located at a 50-meter distance from each other. Collected mosquitoes were stored in silica gel for subsequent morphological identification. 

The use of ZanzaMapp was promoted among the citizens and tourists in Procida in the weeks before and during the entomological monitoring by an advertising campaign in the municipality website and in local newspapers, as well as by hundreds of flyers disseminated in public places (bars, restaurants, shops).

The ZanzaMapp records from 2 September to 16 September 2017 were transformed into a numeric scale from 0 to 3 corresponding to 0, 1–3, 3–30 and >30 mosquito reported, respectively. The transformed ZanzaMapp records were then aggregated over spatio-temporal windows (see below) and compared to HLC data. The spatio-temporal windows were centred on each HLC. Different time-windows were considered in order to find the best balance between choosing a too large, and likely inaccurate, window or a too small one, which would likely imply the need to eventually discard a considerable number of HLC data and ZanzaMapp records. In order to explore how the choice of spatio-temporal windows could affect the relationship between observed HLC values and ZanzaMapp records, we considered for each HLC: (i) 11 spatial (circular) buffers of increasing radius (50 m step increases, from 50 to 500 m) centred on the HLC spot; (ii) 6 time windows starting from 6 days before the HLC and ending the day after sampling completion (as users may enter in the app a record referring to the previous night). For all transformed records falling inside a spatio-temporal window, we computed a spatially weighted mean of aggregated ZanzaMapp records (WM), with the weight (w) computed as a linear function of the radius of the buffer (centred in a HLC spot): *w* = −(1/radius) × distance + 1. Following this transformation, the WM values range from 0 (users record zero abundance) to 1 (very high abundance). HLC data not associated to any ZanzaMapp record within any specific spatio-temporal windows were discarded, implying that the sample database size depends on the choice of the spatio-temporal window considered. Thus, information criteria approaches could not be used to compare models. 

Regression models were created for each spatio-temporal window to investigate the relationship between the predictive performance of aggregated ZanzaMapp records and HLC data. The model was fitted using the Integrated Nested Laplace Approximation approach (INLA) in a Bayesian framework, whose flexibility and computational efficacy allow to extend the model in the future, if needed. The response variable was HLC count observations (y_ij_) at location i and day j and modelled as a Poisson (Y).
yij ~ Poisson (λ_ij_)(5)
log(λ_ij_) = WM_i_ + D_i_ + hour_ij_ + OP_ij_ + ε_j_ + ξ_i_(6)
E (Y) = λ(7)
Var (Y) = λ(8)

Covariates were WM, % of wood (classified from aerial images available from the Italian National Geoportal [18]) at buffer 20 m radius centred on the HLC spot, distance (D) from coastline (m) (computed from coastline available from the Italian National Geoportal [18]), hour of HLC, and a qualitative variable differentiating the technicians (OP). Moreover, the model considered a random effect (ε) for the collection day with Half-Cauchy priors and a spatial field (ξ) modelled using Stochastic Partial Differential Equation (SPDE) with Matérn correlation. For each model, the following data were computed: (i) sample size, (ii) median number of records inside tested spatio-temporal windows, (iii) mean values and credible intervals of all parameter, (iv) Root mean squared error between predicted and observed HLC values in the test set.

## 3. Results

### 3.1. Citizen Interest and Commitment

The numbers and temporal distributions of the 13,669 individual devices which downloaded ZanzaMapp and provided at least one record (10,669, 2206, and 794 in 2016, 2017, and 2018, respectively; hereafter referred to as new users) reflect the intensity of the advertisement efforts, with main peaks associated to major advertising events on national TV and radio programmes (blue bars in Figure 1a). In the first week after the release of ZanzaMapp (week 18) and the three TV interviews (week 19, 2016), 4194 new users were recruited. Afterwards, the numbers declined to ~100/week at the end of August and were negligible during the overwintering season (i.e., after October 2016 until the end of May 2017). In late spring/summer 2017 (in the absence of a specific promotion of ZanzaMapp in the media), new users were <100/week until mid of September, when their rapidly increased up to 811 in the second week of October. This is likely to be associated to the Chikungunya virus outbreak, which caused almost 500 infected cases in Lazio in September–October 2017 [4] and which was largely covered by the media, creating serious concern in the population: This created the opportunity to promote again ZanzaMapp in TV and newspaper interviews. In 2018, when no advertisement was carried out, new users were <50/week. Of the overall users, 55 recruited in 2016 and 167 recruited in 2017 were still active in 2018.

A total of 36,867 records were obtained for the Italian territory from May 2016 (first record sent 4 May 2016 09:09:50 utc) to end of 2018 (last record sent 31 December 2018 22:10:32 utc). Of these, 1968 were discarded due to default coordinates, resulting in 34,899 records considered for subsequent analyses (hereafter referred as records). As in the case of new users, the weekly distribution of records in the 3 years is highly uneven, reflecting again the intensity of the advertising campaign, but also mosquito seasonality, and indicating that users were more motivated to use the app when their perception of mosquito abundance (and nuisance) was higher (multicolour bars in Figure 1a). The highest number of records (~5500) was obtained in the first week following ZanzaMapp release. Afterwards, records were >1000/week until end of July, >500/week until mid-September and decreased down to <50/week until the end of 2016. In 2017, records increased to ~150/week in the summer months until the second half of September, when they increased to 1366/week in late September/early October. From November 2017 to March 2018 (when at least the majority of *Ae. albopictus* is overwintering) records were <50/week (with no records in January and February 2017), and they increased to 200/week in second half of August. 

Of the 13,669 users, 64.6% submitted 1 record only, 15.2% of them 2 records, 16.7% 3–10 records, and 3.5% >10 records. In the case of recurrent users (i.e., users that submitted >1 record), the median and mean number of days between records was ~3 and 12 days (with highly skewed intervals between records, i.e., 25%, 50%, and 75% quantile of intervals between successive records <17 h, ~3 and 8.1 days, respectively, with 95% of intervals shorter than 43 days). This pattern is consistent with the temporal dynamics of records shown in Figure 1a. 

With reference to the time of the observations, 9.2%, 21.8%, 22.9%, and 46.2% of the records were obtained in the morning, in the afternoon, in the night, or in the last hour, respectively, with patterns changing among years, whose biological significance is not straightforward to interpret (Appendix A). 

The geographic distribution of the records was nationwide (Figure 1b), with a median and mean distance between records of ~1.5 km and 27.6 km, respectively (25%, 50%, and 75% quantile distance between records of 43 m, 1.5 km and 11.4 km, respectively, with 95% of intervals <120 km). A linear association is observed between ZanzaMapp records and population densities at log10 scale (where a 10-fold increase in population density correspond in a 19.7 increase in the number of records; Appendix A), confirming the expectation that the highest number of records would come from highly populated areas. The majority of records are from the Lazio Region (19.6%, 45.2%, and 34.2% in 2016 (total = 25,578), 2017 (total = 6187) and 2018 (total = 3134), respectively), reflecting the location of most of the people involved in the studies, Rome, and, in 2017, the increased interest in the app elicited by the chikungunya outbreak in the region [4].

Overall, results suggest a high success of ZanzaMapp in attracting potential users (as shown by the high numbers of new users following major TV promotion events) and a limited one in eliciting long-lasting commitment. The latter one, however, was not to be expected, as ZanzaMapp 1.1.1 did not provide the users with any feedback, incentive and/or notification (e.g., a connection with public health administrations, which could guarantee mosquito control interventions in areas with high numbers of high abundance records). These are known to be the key factors to ensure a long-lasting use of citizen science-based app/projects over time [22,23]. Interestingly, however, the not negligible number of citizens sending >10 records (472) and the fact that they were scattered across Italy (and not concentrated in Rome, where the authors of the paper encouraged colleagues, students and acquaintances to use the app) suggest an interest from a potentially high number of citizen in contributing to mosquito monitoring efforts. 

The above data refer to records associated with the two compulsory questions in ZanzaMapp (i.e., the number of mosquitoes observed and the time of the observation) which precede the record geo-localization on the map and the possibility for users to consult the map and see their own records as well as records from other users. The commitment of users was further assessed by looking at the proportions of records including feedback on the two optional questions, i.e., 77.5% (26,363/34,899) of records including feedback on the location of the observation (52% indoors vs. 48% outdoors) and 60.4% (12,889/21,329, following inclusion of this question in the app on 15-06-2016) including feedback on mosquito biting activity (no bites vs few bites vs several bites). This suggests that the majority of users felt engaged in completing the questionnaire but raises the question whether it would have been better to make the two questions compulsory before the geo-localization and the saving of the records. Indeed, this depends on the relevance of the questions (which in this case were aimed at differentiating between diurnal and nocturnal mosquito species and at assessing the nuisance level, respectively), and on the significance of the feedback obtained (see below).

### 3.2. Mosquito Abundance Records and Biological Significance

The analysis of the records of mosquito abundance obtained depicts scenarios that are overall consistent with the known biology and population dynamics of *Ae. albopictus* and *Cx pipiens* (i.e., the most widespread species in Italy, particularly in urban areas from where most records were obtained), although some critical elements/biases are noticeable.

The results of the spatial analysis for the whole set of data (Appendix A) show that (i) the no mosquito category is more frequent than the other categories at higher elevations, reflecting the less permissive eco-climatic situation beyond 500 m [24,25]; (ii) the high abundance categories are more frequent in areas without full artificial surface cover, in agreement with recognised hot-spots in small green areas within a highly urbanized environment in Italy [6,26]; (iii) the high abundance categories are more frequent in areas closer to the coastline; notably, although no entomological data are available to confirm this results, it is noteworthy that all outbreaks of chikungunya in Italy started in coastal sites [4]; (iv) lack of clear distribution patterns related to distance from inland water bodies.

The results of the temporal analysis show first of all a clear seasonal pattern (with values >75% from week 20 to 43) in the proportion of the different categories of mosquito abundance recorded (0, 1–3, 3–30 and >30 mosquitoes) (Figure 2). The patterns of the two highest abundance categories (>30? mosquitoes) closely reflect expectations based on known mosquito seasonality in Italy, while the pattern of the low abundance categories (1–3 mosquitoes) show high frequencies also in the winter months (with peaks >50% in November 2017–March 2018). The latter results are consistent with previous reports of overwintering adults of the two most widespread urban species, *Ae. albopictus* and *Cx pipiens*, in Italy [27,28]. However, it should be noted that a bias towards positive records is to be expected, as citizens are more motivated to send a record when they see a mosquito or are bitten by it. This general bias is likely to be stronger in the winter months, when users can completely forget about ZanzaMapp unless they unexpectedly see a mosquito and are more motivated to send a record by the unexpected encounter, than in warmer months when mosquito presence is the rule.

Second, the GAM implemented to test the relationship between records of low (≤3) and high (>3) perceived mosquito abundance in 2017 and 2018 and human population density, temperature and hour of the day revealed interesting features (Table 1). The probability of recording a high mosquito abundance followed a similar dynamic in 2017 and 2018, increasing with temperatures up to 29 °C and decreasing at higher temperatures (Figure 3). This is consistent with the biology and seasonal dynamics of the main mosquito species in Italy [29,30]. In the summer period, high abundance records are higher in the morning (between 9:00 and 11:00) and in the afternoon-evening (between 17:00 and 22:00) consistent with the biting patterns of *Ae. albopictus* [31] and *Cx pipiens* [32]. This relationship is not observed during the 2017–2018 winter period (November-March). Interestingly, the probability of recording high mosquito abundance negatively correlates to human population density, consistently with the fact that, at a given absolute density of mosquito, the mosquito/host rate decreases at higher human density. This draws attention to a (largely neglected) bias in the estimates of human-mosquito contact (and consequent risk of arbovirus transmission) based on estimates of mosquito abundance obtained by entomological monitoring only and highlights the possible value of the citizen science approaches to correct this bias. 

Notably, the testing of GAM performance to evaluate model accuracy in predicting high-mosquito abundance based on available data (human population density, temperature, and hour of the day) confirms that the model underestimates user’s records of high mosquito abundance. Specifically, in the train test, the model correctly classified the occurrence of mosquito resulting in a good (i.e., 70.1%) match between fitted and observed values (fitted values classified positive when >0.5). However, the model produced a 25% of false negative predictions (fitted value <0.5 but observed positive record). The same pattern was observed in the test set with the GAM correctly predicting 69.5% of observations (95% CI 67.4–71.8) but showing a 26.2% of false negatives. In other words, the model underestimates high-mosquito abundance, as reported by ZanzaMapp users. This is relevant particularly with reference to the potential use of model estimates to infer mosquito abundance.

Third, the relationship between the proportion of ZanzaMapp records of no, low, high biting activity and records of no, (0), low (1–3), high (3–30) very high (>30) mosquito abundance shows a strong consistency between a prevalence of no-bites records and records of lack of mosquitoes and between a prevalence of high biting activity and records of very high mosquito abundance (Figure 4). This strengthens the quality of the reports and the significance of the possibility of exploiting them to assess mosquito abundance and mosquito/human contacts.

Finally, we compared the population dynamics of the mosquito records reported from outdoors in the morning/afternoon (likely mostly associated to *Ae. albopictus*) versus those reported from indoors in the night hours (likely mostly associated to *Cx pipiens*). The results (not shown) are not consistent with the expectation to see different dynamics reflecting the earlier start of the reproductive season for *Cx pipiens* compared to *Ae. albopictus*. This suggests that, in the current form, ZanzaMapp does not discriminate between an exophilic/diurnal and endophilic/nocturnal species, possibly due to the low number of night hour-records.

### 3.3. Pilot Field Test Validation in Procida Island.

Human Landing Catches (HLCs) and ZanzaMapp promotion were implemented in September 2016 in the island of Procida (Figure 5A). Overall, 191 ZanzaMapp records from 75 users from the island were obtained. Of these, 15.7% were recorded during night hours and were discarded from the subsequent analyses, as they likely referred to nocturnal mosquito species (e.g., *Cx pipiens*) rather than to *Ae. albopictus*, which was the only species collected during the diurnal HLCs carried out in the same week, yielding a total of 764 females and 52 males.

In order to be able to validate the results obtained by ZanzaMapp as a method to estimate *Ae. albopictus* abundance/nuisance, we first explored how the choice of different spatio-temporal windows could affect the relationship between the observed HLC values and the ZanzaMapp records. The results showed a positive relationship between the weighted mean and HLC for some spatio-temporal windows. Shortest (i.e., 1 day or 50 m) and longest (i.e., >3 days or >150 m) spatial/temporal windows provided lower or not significant correlations (Appendix A). This pattern is probably due to the scarcity of data (in the shortest spatio-temporal windows), as well as to possible heterogeneities in mosquito distribution or to the inclusion of records which were temporally or spatially unrelated to the HLC observations (given the wider spatio-temporal windows). These findings suggest that the most appropriate way to obtain valuable information regarding mosquito abundance from ZanzaMapp records is to consider records sent within 100 m and in the 2–3 days preceding the HLCs. Figure 5B shows the estimated abundance of *Ae. albopictus* adults in Procida island based on a 100 m-3 days spatio-temporal window.

## 4. Discussion

A few citizen-science projects have been developed over the last years to complement and/or replace laborious and expensive entomological monitoring and ultimately contribute to prevent the risk of mosquito borne diseases [9]. In particular, two of the most successful ones, MosquitoAlert in Spain and Mückenatlas in Germany [13], focus on monitoring the spread of invasive *Ae. albopictus*—which is still in a process of expansion in these Countries—and to identify other possible invasive *Aedes* species. ZanzaMapp was developed in Italy—where *Ae. albopictus* and *Cx pipiens* are the most common species in urban areas—with the alternative/complementary goal to monitor citizen’s perception of mosquito abundance and nuisance. This means that while MosquitoAlert and Mückenatlas goals are to obtain validated records of the invasive species (by photos and biological samples, respectively), the objective of ZanzaMapp is to obtain as many reports of mosquito presence/nuisance as possible. This was pursued by asking citizens to fill in a very simple and fast questionnaire which was expected to increase the number of records but did not allow validated species identification. Therefore, for their inherent nature, ZanzaMapp data are affected by several biases, such as differences in individual perception, mistakes in the recording procedures, misidentification of other insects as mosquitoes, limits in differentiating between day- and night-biting species. Nevertheless, the results highlight a consistency between the patterns obtained by the analysis of ZanzaMapp records and the known spatio-temporal dynamics of the main mosquito in Italy. Given that the highest number of records is shown to come from highly populated areas, it is reasonable to assume that the results obtained mostly refer to *Ae. albopictus* and *Cx pipiens*, although this may not be the case in some areas. Moreover, the results from the pilot experiments conducted in Procida island show that a relationship exists between ZanzaMapp records and the number of host-seeking *Ae. albopictus*. Further studies are needed to confirm this on a wider geographical scale.

While the high numbers of downloads and records obtained with a very limited advertising effort support the potential for a widespread use of ZanzaMapp, the results showed that most users did not felt engaged in an actual monitoring activity over time. This was not unexpected, given that ZanzaMapp 1.1.1 did not provide any feedback, nor rewards to users. A possible relevant feedback could be represented by maps of mosquito abundance/nuisance produced on the basis of ZanzaMapp records and made available in real-time to citizen and administrations. Figure 6 shows, an example, four maps based on ZanzaMapp data collected in Italy (a), Lazio Region (b), and Rome urban area (c–d). Although very rough and subject to several potential biases (as highlighted above), the former two maps (a,b) highlight the potential value of a citizen science-based approach to produce an overview of mosquito perceived abundance on a large-geographical scale, which is impossible to obtain with entomological monitoring approaches. On the other hand, the smaller-scale maps (c,d) clearly show the potential of the exploitation of citizen records by local public administrations in order to focus control interventions on areas where mosquito nuisance is more acutely perceived. Figure 7 shows two maps (a,b) based on ZanzaMapp data collected in the Lazio region during two consecutive weeks in October 2017, in the middle of the chikungunya outbreak which caused over 400 hundred human infections in Lazio, especially in the city of Anzio [4,33]. Interestingly, the number of records considerably increased from a to b, possibly due to increased concern of the population for the outbreak and to actual increases in nuisance expected in mid-October in the region [6]. Although anecdotic, it is interesting to note that the maps reflect a high variability in mosquito abundance among Municipalities and that the Municipality of Anzio (i.e., the core of chikungunya outbreak [34]) is highlighted as highly infested. Figure 7c shows the temporal profile of the percentage of positive records (≥1 mosquito) reported in the province of Rome in 2016–2018, which could support the optimization of the scheduling of mosquito control interventions.

We believe that a large community participation could be gathered in the future, should ZanzaMapp (or a similar app) be endorsed and properly advertised by public administrations (e.g., Municipalities). A higher number of records will indeed make it possible to produce real-time maps similar to those shown in Figure 7, but with higher spatial resolution, which could complement the entomological data (if any) and eventually be used to identify hot-spot areas on which to target mosquito control interventions, thus improving their effectiveness with modest added costs. Indeed, economical and operational constraints in entomological monitoring make an information-based and spatially targeted control of mosquitoes such as *Ae. albopictus* in public urban areas very rarely feasible, despite the distribution of the species being known as uneven [6]. A more focused and cost-effective mosquito control would indeed represent an incentive for citizen, who would feel directly involved in the monitoring process and in improving the intervention’ effectiveness. Of course, other incentives (such as a direct feedback on mosquito control activities by public administrations via the app itself) could be planned to increase ZanzaMapp usage. A very positive side-effect of the involvement of civil society in the effort to monitor and eventually better control mosquitoes through mobile phone apps is also the potential for massive education and awareness, which can lead to new models of open government and innovation [35]. Indeed, educational material and news in the mosquito field are available both on the ZanzaMapp.it website and in the app. Continuous updates of these sections represent an important instrument to increase user’s affection for the app and promote its use.

## 5. Conclusions

In conclusion, the results obtained with very limited budget by the joint efforts of GH private company and a group of researchers (mostly with no permanent position in the Italian academy, nor in the public sector) confirm the high potential for scaling-up innovative citizen science-based approaches as tools for improved mosquito control. Although we did not carry out an analysis of the costs associated with the running of ZanzaMapp and data analysis vs those associated to classical entomological monitoring, it is important to note that costs, in the case of ZanzaMapp, will be very low and mostly associated to non-recurring investments in technology, to community engagement, and to data analysis. In the case of Mosquito Alert (which also required specialized entomologists for mosquito identification), these costs were estimated to be 7.6-fold lower than standard entomological monitoring [9]. Synergies between scientists, public health institutions, and public administrations in Italy are now needed to promote the scaling-up of the approach as a complementary tool to entomological monitoring or as the only one in case of lack of resources for the latter.

Beyond the national scale, as highlighted by Bartumeus et al. [36] “the challenge is to exploit the inherently scalable nature of Internet networked citizen science to offer an open global toolkit that can aid in the fight against mosquito at the global scale”. This would imply the generation of data and method standards across different projects, as well as the development of a modular project where code and data can be easily shared and reused in other countries, thus facilitating structural interoperability and shared global knowledge. ZanzaMapp could thus evolve as a module focused on the perception of mosquito abundance/nuisance of this global toolkit, as already discussed and agreed during the 2017 workshop sponsored by UN Environment and organized by the Wilson International Center for Scholars’ Science and Technology Innovation Program (STIP), and the European Citizen Science Association (ECSA) in the frame of which the Global Mosquito Alert Consortium was launched, with the participation of representatives of the most successful mosquito pilot projects exploiting apps [11]. This meeting marked the start of the Global Mosquito Alert initiative aiming at contributing to the global surveillance of mosquito species by pooling different “modules” with different goals, ranging from the provision of records of spatial distribution of mosquito species and breeding sites based on image validation by expert entomologists to real-time information on mosquito nuisance/abundance by citizens in a single app.

## Figures and Tables

**Figure 1 ijerph-17-07872-f001:**
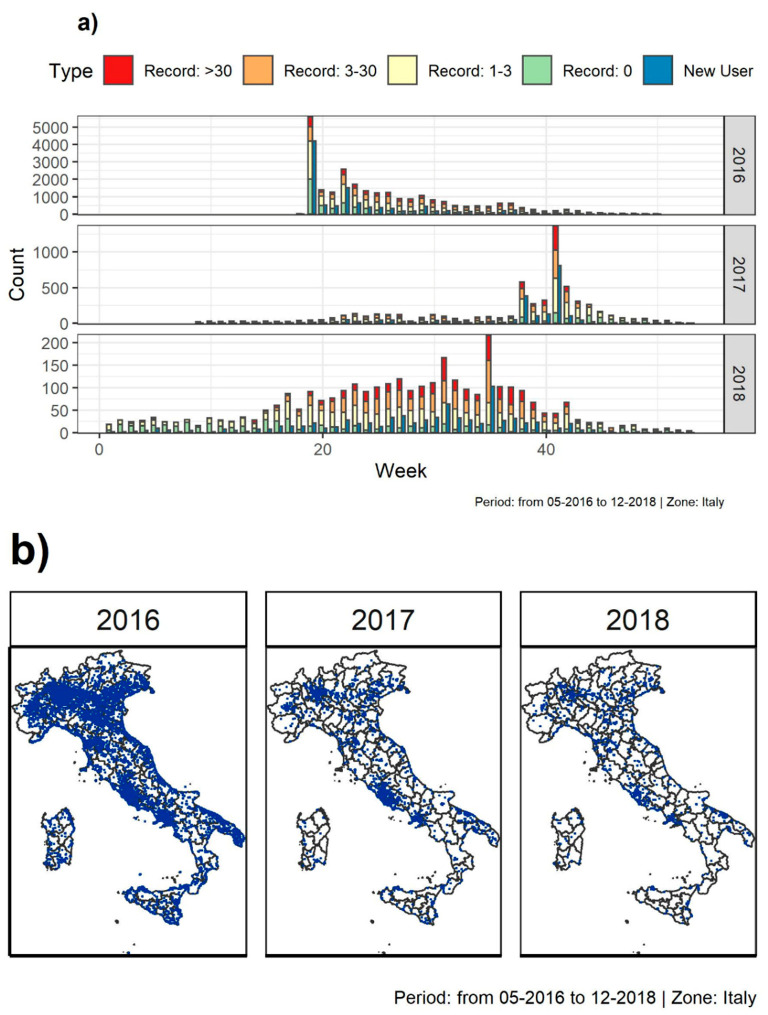
ZanzaMapp records from 2016 to 2018. (**a**) weekly distribution of mosquito abundance categories (0, 1–3, 3–30, >30, multicolour bars) and new users (blue bars). Range of y-axis varies between panels referring to different years in order to help interpretation; (**b**) spatial distribution.

**Figure 2 ijerph-17-07872-f002:**
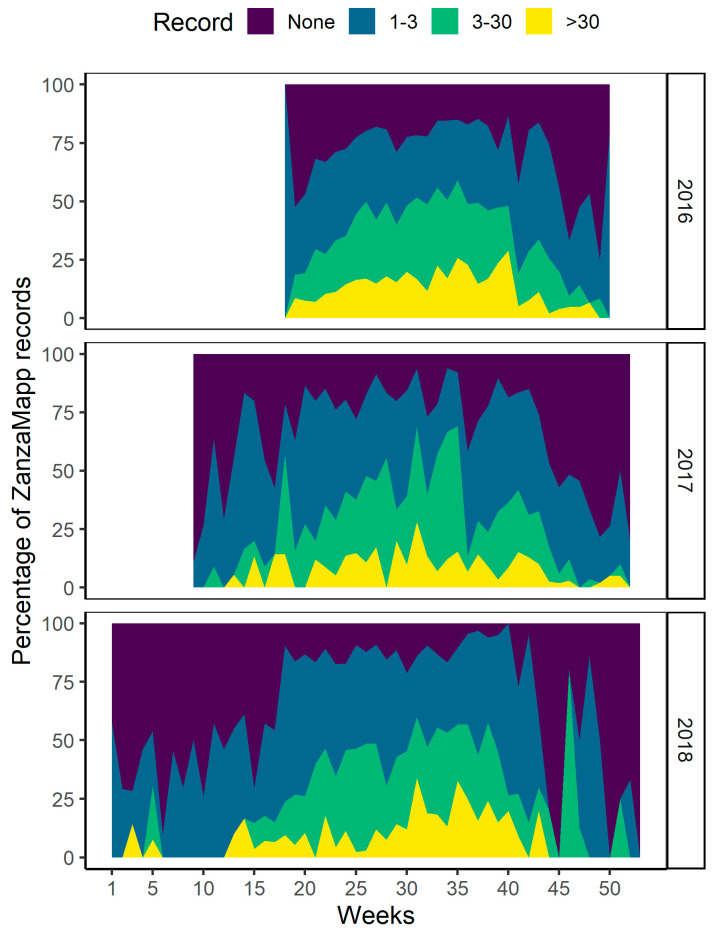
Seasonal distribution of proportion of ZanzaMapp mosquito abundance categories. Dark blue = zero mosquito/week (m/w); light blue = 1–3 m/w; green = 3–30 m/w; yellow = >30 m/w; Gap between week 51 of 2016 and week 8 of 2017 is due to lack of records.

**Figure 3 ijerph-17-07872-f003:**
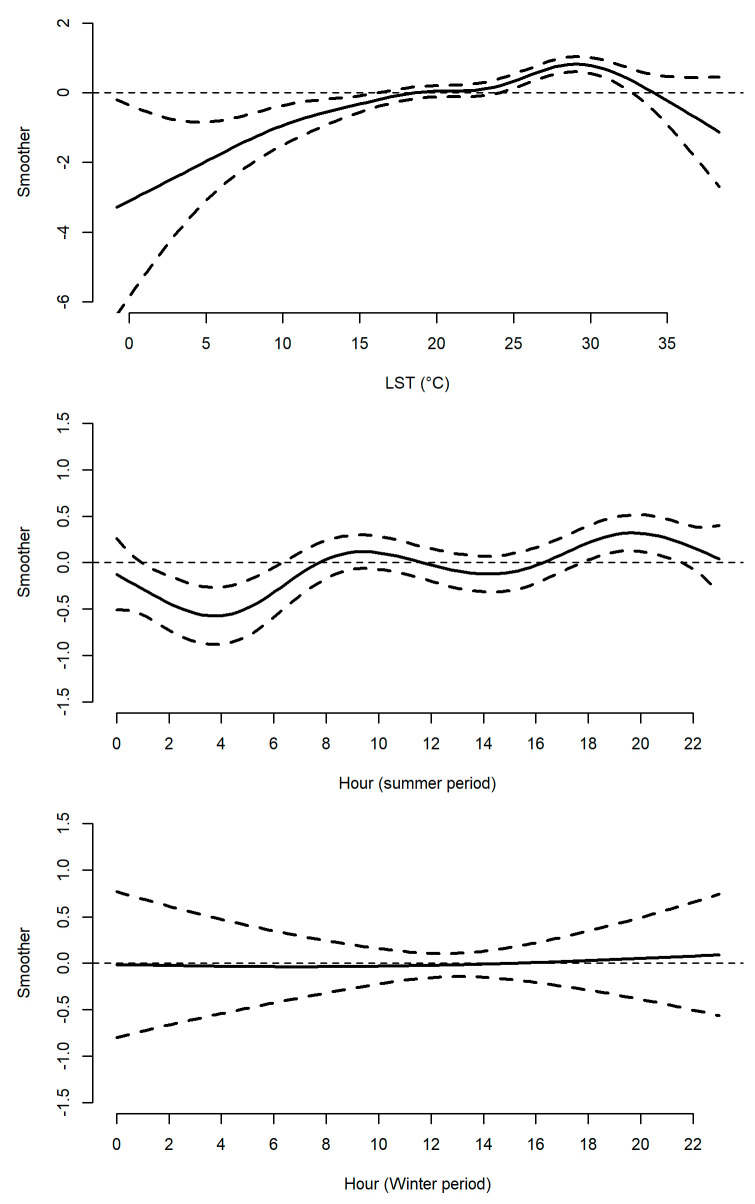
Thin plate splines (y-axis: smoothers) from GAM testing the relationship between ZanzaMapp records of low (≤3) and high (>3) mosquito abundance and human population density, temperature, and hour of the day in 2017 and 2018. LST = land surface temperature. Solid lines represent the mean fitted values, dashed lines represent the 95% confidence intervals.

**Figure 4 ijerph-17-07872-f004:**
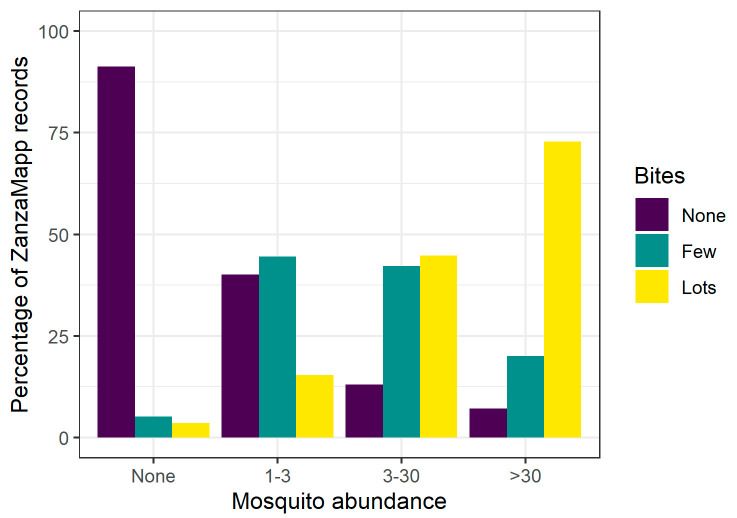
Proportion of ZanzaMapp records of no, low, high biting activity within the 4 mosquito abundance categories (0, 1–3, 3–30, >30).

**Figure 5 ijerph-17-07872-f005:**
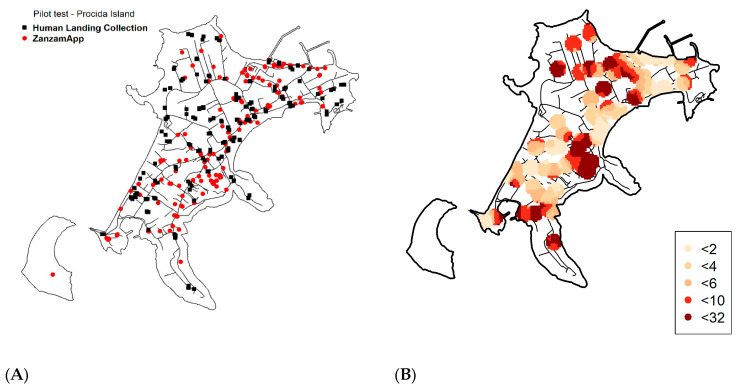
Procida Island (Gulf of Naples, Italy). (**A**) Location of ZanzaMapp records (red circles) and Human Landing Collections (HLC; black squares) in September 2016. (**B**) *Aedes albopictus* abundance estimate based on ZanzaMapp records. Black lines represent roads.

**Figure 6 ijerph-17-07872-f006:**
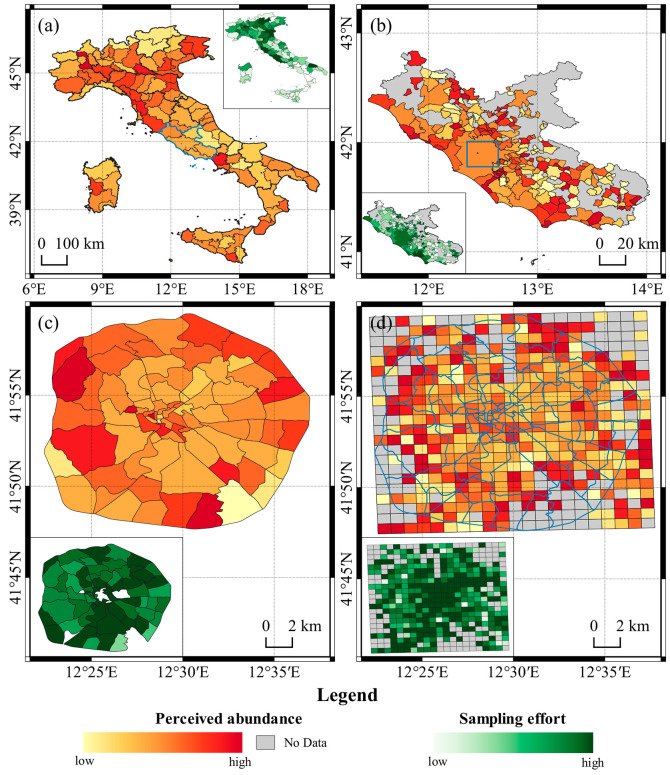
Scales of perceived mosquito abundance based on ZanzaMapp records (May 2016–December 2018) for Italy (**a**), Lazio region (**b**), administrative units in Rome urban area (**c**), and 1 km-spatial grid in Rome urban area (**d**). Inner squares represent maps of sampling effort, i.e., the ratio between number of ZanzaMapp records and human population.

**Figure 7 ijerph-17-07872-f007:**
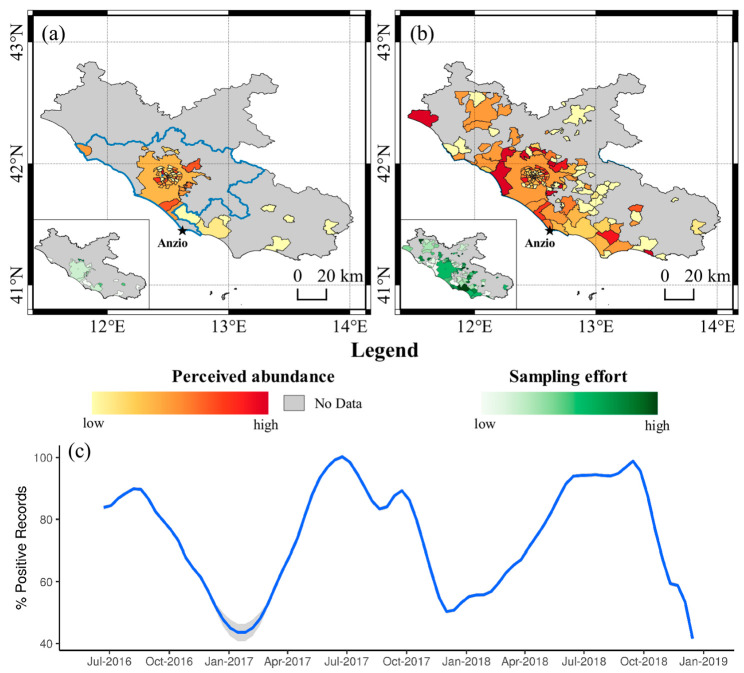
Scales of perceived mosquito abundance based on ZanzaMapp records aggregated for the Lazio region for the weeks 1 October 2017–7 October 2017 (**a**) and 8 October 2017–14 October 2017 (**b**). Inner squares represent maps of sampling effort, i.e., the ratio between number of ZanzaMapp records and human population. (**c**). Temporal profile of the percentage of positive records (≥1 mosquito) reported in the province of Rome in May 2016–December 2018. The grey area from January 2017–April 2017 refers to interpolated data, due to lack of ZanzaMapp records.

**Table 1 ijerph-17-07872-t001:** Result of Generalized Additive Model (GAM) testing the relationship between ZanzaMapp records of low (≤3) and high (>3) mosquito abundance and human population density, temperature, and hour of the day in 2017 and 2018. Human population density was standardized (mean = 376.282, standard deviation = 375.856).

Parameter	Estimate	Std. Error	Z-Value	Pr (>|z|)
Intercept	−0.839	0.065	−12.895	<0.001
Year (2018)	−0.016	0.090	−0.177	0.860
Period (Winter)	−0.816	0.232	−3.516	<0.001
Population	−0.197	0.040	−4.890	<0.001

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
