# Peer review of "ZanzaMapp: A Scalable Citizen Science Tool to Monitor Perception of Mosquito Abundance and Nuisance in Italy and Beyond"

_ijerph, 2020, doi:10.3390/ijerph17217872_

Round 1
Reviewer 1 Report
This study by Caputa & Manica et al. present and evaluate a new smartphone app to evaluate the mosquito abundance and nuisance through citizens. The work is very interesting and well presented. I only have minor comments. I would recommend a clearer distinction between the results and discussions, i.e. a lot of discussion already takes place in the results section. Specific comments: # l. 41: What do you mean by "host pathogen"?# l. 42: "arboviroses" is not a pathogen, but the disease --> rephrase
# l. 51-52: please mention all(!) ZIKV and DENV outbreaks in Europe
# l. 63-64: What do you mean by massive Internet connections?
# l. 64: Minimize which problem?
# l. 79: "prioritizes the number of records" --> unclear
# l. 93, l. 477-478: italize species name
# l.142: "frequency distribution" --> do you mean percentage?
# l. 160-162: I do not understand this sentence. This advantage is specific only for GRASS GIS?
# l. 222: "for 5 minutes"
# l. 248-252: What is the rational behind the selection of the model?
# l. 352-392: There should be a clear seperation between "results" and "discussion"
# l. 462: "Mückenatlas" is not a mobile application. Citizens can sent specimens for identification.
# l. 465: "Cx. pipiens"
Reviewer 2 Report
This is a very interesting manuscript and it is novel for its incorporation of handheld telephones for gathering data on mosquito abundance. This is an important contribution and describes a good way to increase data collection.
The authors must do two things before the manuscript can be accepted.
- Make sure all species names are in italic type; in some places they are not.
- Have a native English speaker review the paper for grammar. I noticed more than a few times where the grammar was somewhat stilted or number was incorrect. For example, more than once I read "citizen" where "citizens" should have been used. Another example is the use of the construction "differently to" - this is a little awkward and there are better ways to express the same thought.
Reviewer 3 Report
With the spread of potential vectors of arboviruses being found more frequently across Europe, and the current autochthonous dengue outbreak of dengue in Italy, this manuscript is very timely. Caputo et al present results from an app to allow mass collection of information about mosquito presence and abundance from a considerable effort of citizen science.
The Introduction is closely tailored to the present study and very informative, describing the state of the art and the specifics of the new ZanzaMapp. The authors achieved a good, if short lived and fluctuating, uptake of the app, and collected some data which they use to produce some useful graphical outputs and correlate with entomological collections to validate the approach. The possible benefits of this over other approaches are described well, but I would like to see a bit more explanation of how the collected data might be used in the Italian context. This is an interesting tool and a valuable study, which has made for an interesting and useful paper. I just have a few suggestions and questions, which are mostly to clarify and explain in a bit more detail the methods and the implications of some of the choices made during the experiment. I am excited to see how this tool and others can continue to be of benefit as part of the ongoing Global Mosquito Alert initiative.
Specific comments:
I am interested in the reasons for the first 2 questions in the app being compulsory and the second 2 optional, and would like to see an explanation for this choice in the Methods section. This is discussed in the Results section so it is not essential to add it to the Methods, but it might be helpful.
Lines 144-146 – I am not sure what the authors mean by ‘imperviousness’, presumably something to do with the ground, perhaps porosity? If so, the analysis has taken presence of water and elevation into account in assessing the environment where records were made, as well as temperature, season and human population density. Some measure of land use or vegetation cover would have been useful – these might be captured to an extent by human density and ‘imperviousness’, but it would be good to discuss the choice and limitations of environmental observations.
I find the description of the data analysis in section 2.3 very clear, except for this sentence, which could do with more explanation: ‘The perceived mosquito abundance is computed as fraction abundance (ranging from 0 to 1) for each polygon by averaging the numeric transformed reports in a specific time interval.’ I am not clear what ‘fraction abundance’ is.
This part is also not clear to me: ‘HLC data not associated to any ZanzaMapp record within any specific spatio-temporal windows were discarded, implying that the sample database size depends from the choice of the spatio-temporal window considered. Thus, information criteria approaches could not be used to compare models.’ As I understand it, the purpose of this exercise is to correlate reported abundance by the ZanzaMapp and HLC, but this sentence seems to suggest that where there were no records in a spatio-temporal window where an HLC was performed, the HLC was discounted, which would bias the results and miss false negative results. Unless citizens were equally likely to report 0 mosquitoes as they were to report 1 or more – this seems unlikely, was this study able to test this? The fact that new users declined in the winter suggests that people were more likely to report mosquitoes when they saw them than to report the absence of mosquitoes, as discussed in the Results section, though the number of reports of no mosquitoes at high altitudes suggests the opposite.
Lines 305-307 – Although there is a reference to Figure S2, it would be nice to indicate something about the ‘changing patterns among the years’.
Figure 3 – more explanation of the word ‘smoothers’ used as the Y axis title would be helpful to those not familiar with GAM testing. Similarly, lines 402-412 is a bit difficult to follow.
Lines 423-429 - this species comparison is interesting, and it is a shame to raise the question but not show the results. If this paragraph is kept in I would like to see this sentence expanded to give more detail and explanation: ‘Results (not shown) are not consistent with the expectation to see different dynamics reflecting the earlier start of reproductive season for Cx pipiens compared to Ae. albopictus.’
Lines 442-457 – this paragraph includes some repetition or clarification of the method which would be better moved to the Methods section. In the Results section here I would like more explanation of the significance of this result – as I read it, the app is most likely to predict actual mosquito abundance in an area if you consider records within 100m and 2-3 days of the place and time you are interested in. How useful does this make the app, for example in identifying hot spots of mosquitoes to target control efforts, or to identify sites where disease transmission is most likely?
Figure S4 – this is interesting data, and could perhaps be moved to the main body of the paper? The legend is redundant as the categories are labelled on each graph.
A close edit for English would be helpful – although the article is clearly written, there are some word choices and phrasing which could be improved, and some minor errors to correct, for example:
Line 24 - massive Internet connections
Line 32 - citizens’ exploitation
Line 93 and lines 216-7 and others - Ae. albopictus should be in italics Line 96 – Full stop not comma in ‘interventions, Finally, we discuss’
Line 130 - dedicated seminaries among students
Line 132 – ‘Any promotion was carried in 2018.’ Do the authors mean that no promotion was carried out in 2018?
